# SQLAgent: Learning to Explore Before Generating as a Data Engineer

## Abstract

Large Language Models have recently shown impressive capabilities in reasoning and code generation, making them promising tools for natural language interfaces to relational databases. However, existing approaches often fail to generalize in complex, real-world settings due to the highly database-specific nature of SQL reasoning, which requires deep familiarity with unique schemas, ambiguous semantics, and intricate join paths. To address this challenge, we introduce a novel two-stage LLM-based framework that decouples knowledge acquisition from query generation. In the Exploration Stage, the system autonomously constructs a database-specific knowledge base by navigating the schema with a Monte Carlo Tree Search–inspired strategy, generating triplets of schema fragments, executable queries, and natural language descriptions as usage examples. In the Deployment Stage, a dual-agent system leverages the collected knowledge as in-context examples to iteratively retrieve relevant information and generate accurate SQL queries in response to user questions. This design enables the agent to proactively familiarize itself with unseen databases and handle complex, multi-step reasoning. Extensive experiments on large-scale benchmarks demonstrate that our approach significantly improves accuracy over strong baselines, highlighting its effectiveness and generalizability.

## 1 Introduction

The recent advancement of Large Language Models (LLMs) has demonstrated remarkable capabilities in complex reasoning and computational tasks(OpenAI, 2023; Bubeck et al., 2023; Mirchandani et al., 2023; Wu et al., 2023; Meta Fundamental AI Research (FAIR) Diplomacy Team et al., 2022). A significant application of these capabilities lies in processing and interpreting the vast amounts of data that underpin modern society. While data exists in many forms, a substantial portion of high-value information is stored as structured data within relational databases(Verbitski et al., 2017; Yavuz et al., 2018). Consequently, there is a surging interest in leveraging LLMs to interact with this structured data, aiming to democratize data access through natural language. This has given rise to Text-to-SQL (also known as NL2SQL), a key research area focused on automatically translating natural language questions into executable SQL queries(Liu et al., 2025; Katsogiannis-Meimarakis & Koutrika, 2023; Kobayashi et al., 2025; Malekpour et al., 2024; Shi et al., 2025; Deng et al., 2022). The core objective is to bridge the gap between human language and relational databases, empowering users to retrieve and manipulate data without needing to master complex SQL syntax.

Despite this promise, current Text-to-SQL approaches struggle to generalize to complex, real-world scenarios(Lei et al., 2025). While these models perform well on some benchmarks, their accuracy often drops significantly when applied to more complex databases, revealing a significant generalization gap(Pourreza & Rafiei, 2024; Gao et al., 2024; Talaei et al., 2024). This happens because effective Text-to-SQL reasoning is highly database-specific. Unlike general code generation, where a model can produce portable logic like a Python sorting algorithm, a valid SQL query is inextricably tied to the unique schema of a specific database. This deep dependency manifests in several key challenges. A model must handle intricate schemas, as their complexity and structure are entirely unique to each database. It must also resolve ambiguous queries, where the correct interpretation of a phrase like "recent customers" depends on knowing the specific column names and business logic embedded in that particular schema. Furthermore, complex, multi-step reasoning is dictated by the database's specific join paths and relationships.

Without prior familiarity with the database, a generic pre-trained model struggles to navigate the unique structure and meaning within a new database(Lei et al., 2025). In contrast, human experts succeed by first building a deep familiarity with the database's unique schema and relationships. This core insight motivates our approach: a preliminary process designed to build this foundational knowledge before attempting the final translation task. To achieve this goal, we propose a novel LLM-based agent framework that operates in two key stages. First, the framework autonomously explores an unfamiliar database to generate a rich, database-specific knowledge base. Subsequently, this knowledge is provided to the agent as in-context examples(Dong et al., 2024; Rubin et al., 2022; Zhang et al., 2022b). This process guides the generation of the final, complex SQL query.

The Exploration Stage autonomously constructs a structured, database-specific knowledge base. The core objective of this stage is to generate a rich set of usage examples for each key component of the database schema, such as its tables and columns. Each example is formalized as a triplet: a schema sub-structure, a corresponding executable SQL query, and its natural language description. To systematically generate these triplets across the entire schema, we first represent the database as a traversable tree structure, where entities like tables and columns are organized as nodes. This tree provides a map for our exploration. To generate triplets, the agent need explore this tree to find and combine meaningful nodes into valid queries. To this end, We then employ a search strategy inspired by Monte Carlo Tree Search(Kocsis & Szepesvári, 2006; Coulom, 2007), guided by an Agent that acts as the core reasoning engine. The agent intelligently navigates this tree to build and test new queries. It achieves this by selecting a series of actions, such as introducing a join or adding a filter. Each successful exploration path results in a new triplet. These triplets are collected to build our knowledge base, enabling the next stage to better write accurate SQL queries. Overall, this process allows the system to proactively familiarize itself with an unknown database without any manual intervention.

In the Deployment Stage, our goal is to effectively utilize the knowledge from the exploration phase to handle complex user queries. To achieve this, we introduce a dual-agent framework where an InfoAgent and a GenAgent collaborate with distinct roles. The InfoAgent is responsible for retrieving the most relevant knowledge triplets from the previously constructed database, based on the user's question. Subsequently, the GenAgent uses this schema fragment to retrieve associated knowledge triplets stored during the exploration stage. The GenAgent then incorporates these retrieved triplets into its reasoning context, using them as in-context examples to guide the final query generation. If the generated query is unsuccessful, the GenAgent provides feedback to the InfoAgent, prompting a new cycle of information retrieval. The two agents then work together in an iterative loop, continuously refining the SQL query through a cycle of example retrieval, generation, and execution feedback. This collaborative, multi-step design allows our system to deconstruct the problem, leading to high accuracy and reliability.

To validate the effectiveness of our approach, we conducted extensive experiments showing that our method significantly outperforms strong baselines on complex, large-scale benchmarks. Our main contributions are:

1. We propose a novel two-stage LLM-based framework that decouples knowledge acquisition from query generation.

2. We develop an autonomous, agent-driven exploration strategy that constructs a structured, executable knowledge base without requiring manual annotations.

3. We introduce a dual-agent reasoning mechanism that iteratively retrieves and integrates in-context examples to generate accurate and executable SQL queries.

4. We will release the full implementation of our framework to foster reproducibility and future research in this domain.

## 2 RELATED WORK

**LLM-based Text-to-SQL Methods.** Research in Text-to-SQL has progressed from early neural parsers to modern LLM-driven approaches (Shi et al., 2025; Deng et al., 2022). Seminal neural methods introduced techniques like graph-based encoders to leverage database schema (Wang et al., 2020) and constrained decoding to ensure syntactic correctness (Scholak et al., 2021). With the advent of large language models, the field has seen significant advancements. Numerous fine-tuning

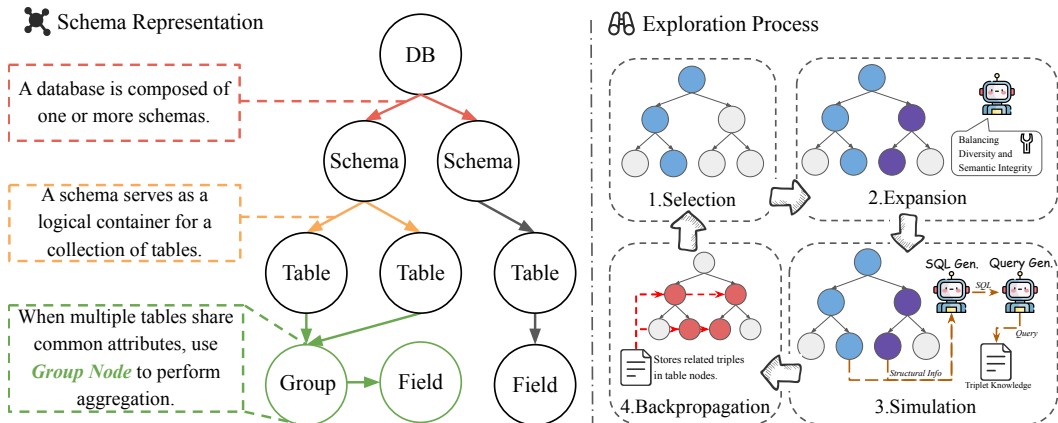

Figure 1: **Databases Representation and Exploration Phase.** The left side of this diagram illustrates our representation and processing of the database structure. The right side of this diagram shows a schematic of our exploration using Tree Search on the existing structure. This includes four phases: selection, expansion, simulation, and backpropagation. This approach enables the collection of a series of triplets.

methods (Li et al., 2024) and advanced LLM-prompting techniques (Dong et al., 2023; Wang et al., 2023a; Zhang et al., 2023; Talaei et al., 2024; Pourreza & Rafiei, 2024; Gao et al., 2024) have achieved strong performance on established benchmarks. However, these approaches still contend with challenges such as limited context windows and adapting to unseen database domains.

**Multi-Agent for Code Generation.** The intersection of generative models and interactive problem-solving has spurred a surge in agent-based frameworks designed to enhance the reasoning capabilities of language models, particularly for code generation (Yao et al., 2023; Zhang et al., 2022a; Chen et al., 2023; Wang et al., 2023b; Shinn et al., 2024; Zhang et al., 2024; Xia et al., 2024). To overcome the limits of a single model, a prominent strategy is to deploy multiple agents in coordinated roles. In this multi-agent paradigm, agents specialize and interact, often through patterns like cooperative decomposition (Li et al., 2023a) or hierarchical task delegation (langgenius, 2023). To make these interactions more robust, other works have focused on designing special action spaces to standardize agent operations (Wang et al., 2024; Yang et al., 2024). Inspired by these successes, our method employs a similar strategy with dedicated explorer and generator agents for text-to-SQL translation.

## 3 METHODOLOGY

Our work addresses the challenge of generalizing Text-to-SQL systems to unfamiliar databases by introducing a novel two-stage framework. First, an **exploration stage** autonomously constructs a database-specific knowledge base without requiring manual supervision. Subsequently, a **deployment stage** employs a dual-agent framework to effectively leverage this acquired knowledge for accurate and robust query synthesis.

### 3.1 DATABASE SCHEMA REPRESENTATION

Our approach to database exploration begins by transforming the conventional relational schema into a traversable, tree-like structure. The database, its tables, and their corresponding fields are treated as distinct nodes in the tree, explicitly capturing their nested relationships. However, a significant challenge arises from the structural redundancy common in large scale databases. For instance, time-based sharding often produces numerous tables with identical schemas, such as daily logs or hourly snapshots. Naively modeling each table independently would require connecting every table node to its respective field nodes, resulting in high representational complexity and redundancy. To address this issue, we introduce an abstraction termed the **Shared Field Group**. This component acts as a canonical, reusable template that encapsulates the common field structure for a set of struc-

turally identical tables. Consequently, instead of each table node maintaining numerous individual connections to its field nodes, it establishes a single link to the appropriate **Shared Field Group**. This design, illustrated in Figure 1, significantly simplifies the overall schema representation and clarifies the inherent relationships among tables with identical structures. By abstracting common structures, this approach reduces the complexity of representing the relationships between tables and fields from $O(N \times M)$ to $O(N + M)$, where $N$ is the number of sharded tables and $M$ is the number of shared fields. This abstraction not only streamlines the schema but also improves the interpretability and efficiency of the subsequent exploration process. The specific definition and identification algorithm for this structure are detailed in Appendix A and Appendix B.

## 3.2 Exploration Stage: LLM-Guided Tree Search

The goal of the exploration stage is to systematically acquire prior knowledge about the target database. This knowledge should capture not only the static schema structure but also how its components can be combined to form meaningful queries. Specifically, we aim to generate a knowledge base where viable structural patterns are translated into executable SQL queries and aligned with their natural language semantics. We formalize the exploration results as a set of triplets $(S, Q, U)$, where $S$ represents a subset of the database schema's structural, $Q$ is a corresponding executable SQL query, and $U$ is the natural language semantically aligned with $Q$.

To manage the immense complexity arising from numerous schema components and their combinatorial possibilities, we propose a novel exploration framework inspired by Monte Carlo Tree Search(Kocsis & Szepesvári, 2006), where we replace its traditional heuristic-based search policy with an LLM that serves as a semantic reasoning engine. The entire exploration process continues until a predefined termination condition is met, such as generating a target number of valid triplets or reaching a maximum number of iterations. This LLM-driven method retains the structured, iterative cycle of exploration, allowing the system to build complex queries step-by-step, guided by the model's understanding rather than numerical rewards alone. The process, illustrated in Figure 1, consists of four distinct phases. These phases are: LLM-guided selection and expansion, simulation, and backpropagation of outcomes.

**LLM-Guided Selection and Expansion.** The goal of this phase is to incrementally construct complex and semantically diverse SQL queries from the database schema. Our method achieves this by leveraging an LLM to perform both selection and expansion in a single, reasoned step. At each node in the search tree, the LLM is prompted with the necessary context to make an decision. This context includes the current query state (a sequence of previous actions) and the available schema context (a simplified JSON structure of reachable tables and columns). The LLM's task is to select the most promising subsequent action from a predefined discrete action space, with options such as `Select Column`, `Add Constraint`, and `Apply Aggregation`. A detailed description of each action is provided in Appendix C. The LLM's choice directly creates a new node, expanding the search tree. The objective of this expansion is to guide its growth toward constructing more sophisticated queries. This strategy ensures the generation of semantically rich examples that cover a wide range of database operations and schema interactions, moving beyond simple single-column retrievals.

**SQL Simulation.** From the newly expanded node, the simulation phase leverages an LLM to generate a complete and executable query. The model is prompted to construct a SQL query that is semantically consistent with the structure represented by the current node. To guide this process toward generating meaningful queries, the LLM is instructed to prioritize using tables and columns that have associated documentation or comments. This metadata provides valuable semantic clues for building a more relevant query. The process operates on a dynamically constrained set of schema nodes, excluding those already incorporated into the current query path (with the exception of key columns like IDs) to ensure broad exploration. The LLM then uses the context from the expanded node to construct a correct SQL query. For instance, if the expanded node contains the columns `age`, `height`, and `gender` from a `users` table, the LLM might complete this structure by generating the SQL query `SELECT * FROM users WHERE age > 20 AND gender = 'Male';` and its corresponding natural language description, "Find all male users older than 20."

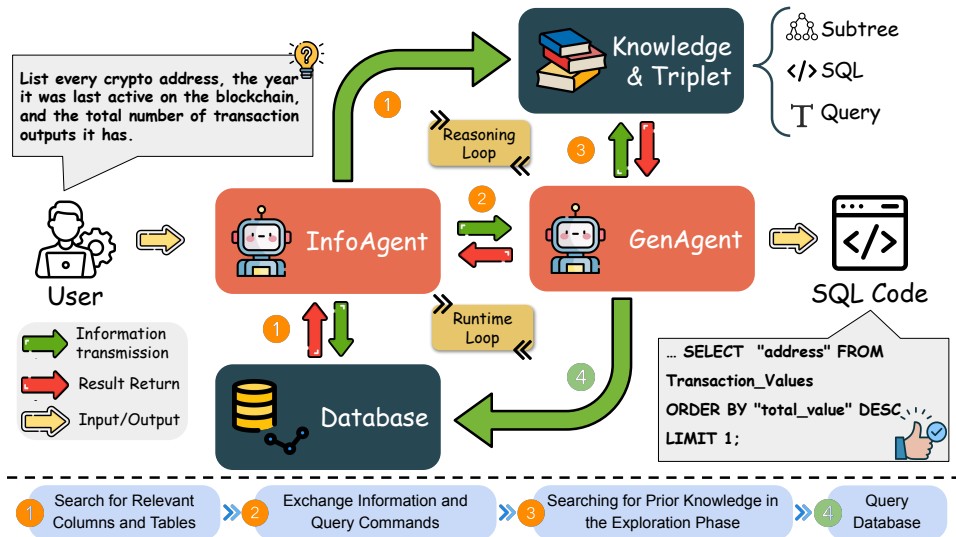

Figure 2: **SQL Deployment Stage.** In this stage, the database information obtained in the exploration stage and the user's actual query are used to generate the SQL. The dual-agent architecture controls the information acquisition context and SQL generation context of the agent, enabling the system to process complex SQL statements while maintaining high query accuracy.

The resulting query $Q$ is then executed against the database. If the query runs successfully and returns a non-empty result, we proceed to generate its natural language counterpart by providing the structural information $S$ and the SQL query $Q$ as input to an LLM, which generates a semantically consistent user query $U$. This culminates in a complete, high-quality triplet $(S, Q, U)$. If the query fails or returns an empty result, the simulation concludes, and this outcome is passed to the next phase.

By first defining a valid, localized schema structure and then generating the corresponding SQL and natural language description from it, our method produces highly accurate and dependable examples. Since each example is grounded in a real data sub-structure, it guarantees high semantic alignment and serves as a reliable reference during the deployment stage.

**Backpropagation.** The primary goal of the backpropagation phase is to systematically record the outcomes of each exploration and use this knowledge to guide deployment stage. To achieve this, the system updates the historical context of each node along an exploration path based on the outcome from the simulation phase. For a successful exploration that yields a valid $(S, Q, U)$ triplet, this triplet is recorded as positive feedback. It is stored on all entity nodes, such as the tables and columns, that were part of the query's creation path. This process enriches each schema component with a history of its successful usage in meaningful combinations. Conversely, if an exploration fails because the query is invalid or returns an empty result, a negative outcome is propagated to the nodes on that path. This feedback serves to lower the selection priority of this path in future iterations. It effectively discourages the model from repeatedly pursuing unproductive combinations. Ultimately, this accumulated historical information is crucial for adaptive learning. In subsequent selection steps, the prompt provided to the LLM is augmented with the feedback stored on the currently visible nodes. By learning directly from the consequences of its past actions, the LLM continuously refines its exploration strategy. It learns to favor paths that lead to valid, non-empty results, creating a self-optimizing exploration process.

## 3.3 DEPLOYMENT STAGE: DUAL-AGENT SQL SYNTHESIS

The SQL deployment phase aims for the robust and accurate translation of natural language into SQL, a task made difficult by the information gap between user queries and complex database schemas. To bridge this gap and effectively utilize the knowledge acquired in the exploration stage, we introduce a dual-agent framework. This framework manages a dynamic, iterative process that

begins by assembling an actionable context from the saved knowledge base, followed by cycles of context refinement and query construction. This process, illustrated in Figure 2, consists of an InfoAgent for schema interaction and context management, and a GenAgent for knowledge-driven SQL synthesis.

The workflow begins when the InfoAgent receives a user's query. Its primary challenge is to bridge the semantic gap between the user's natural language and the rigid, formal structure of the database schema. To address this, the agent first performs *schema grounding*, a process to identify an initial set of relevant tables and columns. It leverages an LLM to analyze the user's question and extract key semantic keywords and entities. To make the schema searchable, we pre-process it by creating a vector embedding for each column. This is done by concatenating a column's name, data type, and any available comments into a descriptive string, which is then converted into a high-dimensional vector using a sentence-embedding model. The InfoAgent then uses the extracted keywords to perform a semantic search against this vector database, retrieving the top-k schema components that are most semantically aligned with the user's intent. This initial retrieval provides a strong, focused starting point for query construction.

However, this initial top-k set may be incomplete, often missing crucial components required for complex operations like multi-table joins or implicit user needs. To ensure the context is comprehensive, the InfoAgent performs a second expansion step. It provides the user's original query and the preliminary set of retrieved schema components as input to another LLM prompt. The LLM is tasked to act as a database expert, analyzing the relationships between the provided components and reasoning about what additional tables or columns are necessary to form a complete, executable query. For instance, if the initial set contains a `user_name` column from a `users` table and an `order_amount` from an `orders` table, the LLM would infer the need to include the `user_id` and `customer_id` keys to facilitate the join. This refined and enriched schema context containing both directly relevant and logically inferred components, is then transmitted to the GenAgent.

Upon receiving the refined schema context from the InfoAgent, the GenAgent initiates the synthesis process. Its core task is to select the most relevant examples from the knowledge base to guide the final query generation. To achieve this with high precision, our retrieval mechanism focuses on the semantic content of the SQL query component $(Q)$ within each stored $(S, Q, U)$ triplet. Specifically, we pre-process the entire knowledge base by vectorizing the SQL query $Q$ of each triplet using a code-embedding model. This transforms our knowledge base into a high-dimensional vector space, where each triplet is represented by its query's semantic vector, enabling efficient similarity searches. When a new user request is processed, the GenAgent uses the user's natural language query and the provided schema context to generate a query embedding that represents the user's intent. This embedding is then used to perform a similarity search against the vectorized knowledge base, retrieving the top-k triplets whose SQL queries are most semantically similar to the target query. These top-k triplets serve as powerful, database-specific in-context examples. The GenAgent then constructs a final, comprehensive prompt by integrating the user's original question, the refined schema context, and these retrieved few-shot examples. Following generation, the query is executed against the database to validate both ability to return a non-empty result.

The outcome of this execution, whether a successful result or an error, triggers a collaborative refinement loop by feeding back to the InfoAgent. The InfoAgent first updates its internal state by recording which schema components were utilized in the generated query. If the query failed due to a syntax error, the InfoAgent refines its strategy for the next attempt. It prunes the context by removing schema components that were provided to the GenAgent but ultimately unused in the failed query, thus narrowing the context window. Concurrently, it re-analyze the user's query for secondary keywords to recall additional, potentially useful tables and columns. Conversely, if the query executed successfully, then performs a final step: a semantic fidelity check, where it assesses the alignment between the query's output and the intent of the user's original natural language request. This iterative cycle continues until either a syntactically valid and semantically correct SQL query is successfully validated, or a predefined maximum number of iterations is reached. The process then concludes by returning the successful result or a failure notification to the user. For a formal algorithmic representation, please refer to Appendix D.

## 4 EXPERIMENTS

This section presents a comprehensive experimental evaluation of our proposed two-stage method. Our primary objective is to assess the effectiveness and efficiency of the SQLAgent. We begin by detailing the experimental setup, followed by a thorough analysis of the results from comparative and ablation studies.

### 4.1 EXPERIMENT SETUP

**Benchmark.** We evaluate the proposed method on the Spider 2.0-Snow benchmark(Lei et al., 2025). This benchmark includes a large number of enterprise-level SQL queries, some exceeding 100 lines in length, representing a significantly higher level of complexity compared to traditional text-to-SQL tasks. Each subtask contains 547 examples spanning over 150 databases, with each database containing approximately 800 columns on average. Consistent with the Lei et al. (2025), we classify SQL query difficulty based on token count: Easy (fewer than 80 tokens), Medium (80-159 tokens), and Hard (160 or more tokens).

**Evaluation Metrics.** We evaluate the performance of our method using the widely adopted Execution Accuracy (EX) metric(Li et al., 2023b; Yu et al., 2018; Lei et al., 2025). Execution Accuracy compares the execution result of the predicted SQL query with that of the ground-truth query on a given database instance. This metric provides a more precise estimate of model performance, as there may be multiple valid SQL queries for a single question. To account for the stochastic nature of large language models (LLMs), we also report PASS@K, which measures whether correct results can be obtained within $K$ runs, thereby evaluating the stability of the outputs under repeated executions. To evaluate the overall efficiency of the model, we measure the number of calls made to the Large Language Model and the number of executions sent to the database. We then report the average of these counts across all tasks in the benchmark.

**Baselines.** Our baseline method, inspired by Deng et al. (2025); Yu et al. (2018), employs a self-refinement mechanism to navigate large database schemas. Initially, we construct a textual index of the schema by concatenating table and column names. When a user submits a query, our system retrieves the Data Definition Language (DDL) of the most relevant tables from this index and provides them as context to a Large Language Mode. If an execution attempt returns an empty result or produces a syntax error, the LLM initiates a refinement loop. It iteratively attempts to correct the SQL query using the execution feedback, without re-querying the database structure. This process continues until a valid query is generated or a predefined stopping condition is met.

### 4.2 EXPERIMENTAL DETAILS.

Throughout our experiments, we retrieve the top-3 most relevant database tables ($k = 3$) to populate the context and set the maximum number of self-refinement attempts to 5. To ensure a fair comparison, these hyperparameter settings are kept consistent for both the baseline and our proposed method. Unless otherwise specified, all experiments utilize GPT-4o as the backbone LLM. The temperature for the all the LLM were set to 0.7. The multi-agent framework is built upon the LangGraph library. Vector embeddings are generated using the `text-embedding-3-small` model from openai, and all vectorization tasks are subsequently handled by the Faiss library. Our data modeling framework is built entirely on Neo4j, and we employ the Cypher query language for all graph traversal.

### 4.3 RESULTS AND ANALYSIS

This section evaluates the proposed two–stage framework, focusing on its effectiveness, efficiency, and robustness. Unless otherwise specified, accuracy is reported using execution accuracy (EX) , while efficiency is assessed by wall-clock time and model token cost.

**Comparative Results.** To evaluate the effectiveness and efficiency of our proposed framework, we compared SQLAgent against two strong baselines, `Spider-Agent` and ReFoRCE, with detailed results presented in Table 1. Our full method significantly outperforms both baselines, achieving an overall Execution Accuracy (EX) of **25.78%**, compared to 20.84% for ReFoRCE and 12.98% for Spider-Agent. The performance gap is particularly pronounced on complex queries; on the "Hard"

Table 1: **Performance Comparison of Different Method.** This table presents a detailed comparison of execution success (EX) and efficiency metrics. EX is reported as a percentage for easy, medium, and hard difficulty levels, along with the overall score. Efficiency is measured by the average number of LLM and Database (DB) calls per query. We evaluate our proposed SQLAgent with different components against other baseline methods and state-of-the-art methods.

| Method | Strategy | EX (%) | | | | Efficiency | |
|---|---|---|---|---|---|---|---|
| | | Easy | Medium | Hard | Overall | LLM Calls | DB Calls |
| Spider-Agent | Agentic | 24.22 | 11.38 | 6.94 | 12.98 | 11 | 3 |
| ReFoRCE | Consensus | 4 9.22 | 17.00 | 5.23 | 20.84 | 3.5 | 3.9 |
| SQLAgent | Baseline | 32.81 | 14.57 | 0.00 | 14.26 | 3.0 | 4.2 |
| | w/ Exp. Stage | 41.40 | **19.03** | 5.81 | 20.10 | 4.1 | 5.2 |
| | full method | **57.81** | 18.62 | **12.21** | **25.78** | 5.2 | 3.6 |

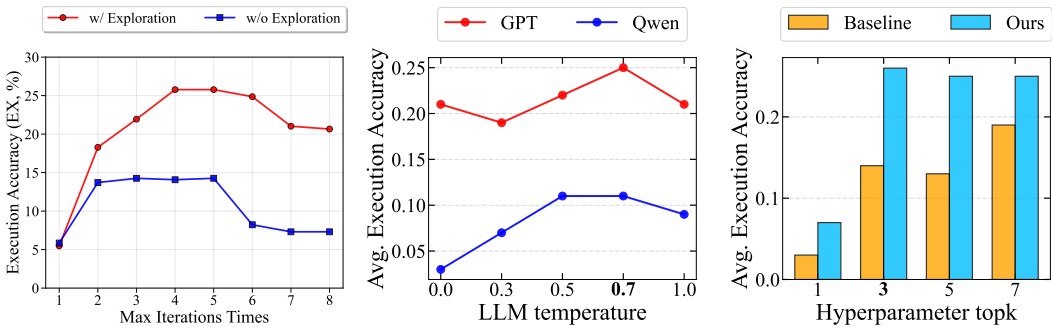

(a) Performance During Iteration.          (b) Analysis of Two Key Hyperparameters

Figure 3: **Analysis of the Exploration Stage and Key Hyperparameters.** The left figure shows Execution Accuracy as a function of agent iterations, demonstrating that the Exploration Stage consistently improves performance. The middle figure presents the effect of LLM temperature on different models, with the GPT model's accuracy peaking at a temperature of 0.7. The right figure evaluates the top-k schema retrieval parameter, showing our method (Ours) outperforms the Baseline across all settings and that its performance gain saturates for k values greater than 3.

subset, our method achieves an EX of **12.21%**, demonstrating a clear improvement over competitors that struggle with these tasks. To isolate the contributions of our framework's components, we conducted an ablation study. Our Baseline model establishes a performance of 14.26% overall EX, failing entirely on hard queries. Incorporating the knowledge from the Exploration Stage (w/ Exp. Stage) improves the overall accuracy to **20.10%**, confirming the value of the proactively constructed knowledge base. Finally, the full method, which integrates the dual-agent framework of the Deployment Stage, achieves the highest accuracy of **25.78%**. This final performance gain validates that the dual-agent architecture effectively utilizes the acquired knowledge for robust SQL synthesis. In terms of efficiency, our full method averages **5.2** LLM calls and **3.6** DB calls, as detailed in Table 1. This configuration is notably more efficient than Spider-Agent, which requires 11 LLM calls. While it uses more LLM calls than ReFoRCE (3.5), it requires fewer database interactions (3.9) to achieve its superior accuracy. The ablation study further clarifies this trade-off: introducing the Exploration Stage modestly increases cost over the baseline. However, the subsequent addition of the dual-agent framework reduces the number of database calls from 5.2 to 3.6. This indicates that the framework's structured reasoning not only enhances accuracy but also leads to fewer unnecessary query executions, validating the overall design.

**Performance During Iteration.** Figure 3a tracks EX as a function of the maximum iteration budget for the dual-agent loop, comparing the performance of our framework with and without the knowledge base from the Exploration Stage. In this analysis, we control the number of collaborative cycles between the InfoAgent and GenAgent; the GenAgent is compelled to generate a final SQL query upon reaching the iteration limit.

The results clearly demonstrate the value of the exploration-derived knowledge. For the model equipped with this knowledge (the red curve), accuracy shows a strong positive correlation with the number of iterations, rising from 5.5% to a peak of 25.8% at four iterations. This trend indicates that with each cycle, the agents effectively leverage the knowledge base to progressively refine the context and retrieve relevant examples, leading to more accurate query construction. In contrast, the model without the knowledge (the blue curve) exhibits limited improvement, with accuracy plateauing at a much lower peak of 14.1%. More significantly, its performance sharply degrades after five iterations. This phenomenon suggests that without validated priors, the iterative refinement process is less stable. The agents struggle to distinguish productive refinement paths from unproductive ones, and an accumulation of redundant or ambiguous feedback can even lead them to revise an initially correct query into an incorrect one.

**Hyperparameter Analysis.** We conduct experiments to determine the optimal settings for two critical hyperparameters: the schema retrieval top-k and the LLM temperature. The $k$ parameter defines the number of most relevant columns retrieved from the database to address a user's query. To analyze its impact, we compare the performance of our full method against the baseline across various $k$ values. As illustrated in the right panel of Figure 3b, the performance of our method rapidly improves and approaches saturation at $k = 3$. Increasing $k$ beyond 3 provides diminishing returns at a higher computational cost for our method. In contrast, the baseline requires a larger $k$ but is constrained by the LLM's context window. We therefore select $k = 3$ as the optimal setting to balance accuracy with efficiency. The temperature parameter controls the randomness of the LLM's output(Peeperkorn et al., 2024). The middle panel of Figure 3b shows the effect of temperature on execution accuracy. We observe that a higher temperature (*e.g.*, 0.7) enables the LLM to generate more varied and effective search terms. Consequently, we set the LLM temperature to 0.7 in our experiments.

**LLM Backbones Analysis.** Table 2 compares the performance of our full method against the baseline across four different Large Language Model (LLM) backbones, reporting the per-query token cost and Execution Accuracy after 8 passes (EX@8)(OpenAI, 2023; Qwen et al., 2025; Li et al., 2023b). The data shows that our framework delivers consistent and substantial improvements regardless of the underlying LLM's capability. As detailed in the Δ column, our method achieves a robust absolute accuracy gain of approximately **+10%** across the more

Table 2: **Performance Comparison Across LLM Backbones.** The table compares our full method against a baseline across various LLMs. We report the per-query cost (k tokens) and Execution Accuracy (EX@8, %). The Δ column shows the absolute improvement of our method.

| LLM | Cost | EX@8 (%) | | Δ |
| --- | --- | --- | --- | --- |
| | | Baseline | Full Method | |
| Qwen2.5-7B | 20.8k | 7.12 | 14.99 | +7.87 |
| GPT-4o | 15.2k | 21.57 | 31.99 | +10.42 |
| GPT-5 | 12.5k | 22.49 | 32.90 | +10.70 |
| Claude-Sonnet-4.0 | 11.6k | 29.97 | 39.12 | +9.15 |

powerful models. The consistent performance improvement shows that our two-stage architecture, which combines proactive knowledge exploration and dual-agent reasoning, acts as a model-agnostic enhancement for any LLM it is paired with. Notably, the impact is particularly pronounced on the smaller model, where our method more than doubles the EX@8 of Qwen2.5-7B-Instruct from 7.12% to **14.99%**. This suggests our structured approach can effectively compensate for the reduced reasoning capacity of smaller models, highlighting its value and scalability.

## 5 CONCLUSION

We introduce SQLAgent, a framework that emulates a human data engineer's problem-solving process. By first autonomously exploring a database to build context-specific knowledge and then leveraging it for deployment, SQLAgent provides a scalable and adaptive solution for large-scale, enterprise-level databases where conventional methods falter. Extensive experiments validate this approach, demonstrating state-of-the-art performance on challenging benchmarks. Ultimately, our work represents a significant step towards creating more autonomous and capable agents for data interaction, advancing the goal of democratizing access to complex structured data.

ETHICS STATEMENT

This research aims to make a positive societal contribution by facilitating access to structured data through natural language. All of our experiments are conducted on public academic benchmarks and do not involve any new personal data collection or human subjects. We recognize the potential for misuse of any data access tool, but our framework is designed to operate within existing data security and access permission frameworks, rather than circumventing any security protocols. In accordance with the principles of research integrity, we explicitly disclose our use of large language models for writing assistance in the appendix. All authors have read and are committed to adhering to the ICLR Code of Ethics.

REPRODUCIBILITY STATEMENT

We are committed to ensuring the reproducibility of this research. We will publicly release the source code for our framework to support subsequent academic research. A detailed description of our model architecture and the two-stage exploration-deployment methodology is presented in Section 3. The complete experimental setup, including the benchmarks, evaluation procedures, and all hyperparameter configurations required for replication, are thoroughly documented in Section 4. Further implementation details and relevant definitions can be found in the appendix.

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

# A   DEFINITION OF DATABASE STRUCTURE

As introduced in the main methodology, we represent the relational database as a traversable graph structure to facilitate autonomous exploration. This model is composed of distinct node and relationship types that capture the hierarchical and relational nature of the database schema. This appendix provides a formal definition of each component, their respective properties, and an overview of the graph's statistical composition.

## A.1   GRAPH SCHEMA OVERVIEW

The graph model consists of five distinct node types and five relationship types that define their connections. The primary node types are:

- **Database**: The root node representing the entire database instance.
- **Schema**: Represents a schema or dataset within the database.
- **Table**: Represents a single table.
- **Field**: Represents a column within a table.
- **Shared Field Group**: Our proposed abstraction for a reusable set of fields shared by multiple tables.

The relationships between these nodes define the structural integrity of the graph. Table 3 outlines the valid connections between node types.

Table 3: Relationship types defining the connections between nodes in the graph schema.

| Start Node Type | Relationship Type | End Node Type |
|---|---|---|
| Database | HAS_SCHEMA | Schema |
| Schema | HAS_TABLE | Table |
| Table | USES_FIELD_GROUP | SharedFieldGroup |
| Table | HAS_UNIQUE_FIELD | Field |
| SharedFieldGroup | HAS_FIELD | Field |

## A.2   NODE PROPERTY DEFINITIONS

Each node in the graph contains a set of properties that store its metadata. The following tables detail the properties for each of the five node types.

Table 4: Properties of the `Database` node.

| Property | Description | Example |
|---|---|---|
| `<id>` | Internal unique identifier for the node. | 15313 |
| name | The name of the database instance. | OPEN_TARGETS_PLATFORM_2 |
| type | Node type specifier. | database |

## A.3   GRAPH COMPOSITION STATISTICS

To provide a sense of scale, Table 9 summarizes the composition of the graph constructed from our experimental datasets. The statistics highlight the prevalence of fields, which constitute over 90% of all nodes.

A key component of our model is the `SharedFieldGroup`, designed to reduce redundancy. Our analysis confirms its effectiveness:

- **Total Groups:** There are **542** unique `SharedFieldGroup` nodes.

Table 5: Properties of the `Schema` node.

| Property | Description | Example |
|---|---|---|
| `<id>` | Internal unique identifier for the node. | `5430` |
| `database` | Name of the parent database. | `NOAA_DATA` |
| `description` | A description of the schema, if available. | `-` |
| `name` | The name of the schema. | `NOAA_SIGNIFICANT_EARTHQUAKES` |
| `type` | Node type specifier. | `schema` |

Table 6: Properties of the `Table` node.

| Property | Description | Example |
|---|---|---|
| `<id>` | Internal unique identifier for the node. | `185` |
| `database` | Name of the parent database. | `COVID19_USA` |
| `ddl_summary` | A summary of the table's DDL statement. | `Table with 245 columns` |
| `fullname` | The fully qualified name of the table. | `CENSUS_BUREAU_ACS_2...` |
| `name` | The name of the table. | `SCHOOLDISTRICTSECONDARY...` |
| `schema` | Name of the parent schema. | `CENSUS_BUREAU_ACS` |
| `type` | Node type specifier. | `table` |

Table 7: Properties of the `SharedFieldGroup` node.

| Property | Description | Example |
|---|---|---|
| `<id>` | Internal unique identifier for the node. | `2079` |
| `database` | Name of the parent database. | `NEW_YORK_PLUS` |
| `description` | Auto-generated description of the group. | `FieldGroup_3ebfe5f9...` |
| `field_count` | Number of fields encapsulated in this group. | `24` |
| `field_hash` | An MD5 hash of the sorted field names, used to identify unique groups. | `3ebfe5f9...` |
| `name` | Auto-generated name for the group. | `FieldGroup_3ebfe5f9` |
| `schema` | Name of the parent schema. | `NEW_YORK_TAXI_TRIPS` |
| `type` | Node type specifier. | `shared_field_group` |

- **Average Fan-out:** On average, each group is linked to **10.6** tables, indicating frequent reuse.
- **Maximum Fan-out:** The most utilized group is shared by **334** distinct tables.

This analysis demonstrates the high prevalence of redundant schema structures in large-scale databases and validates our abstraction's utility in creating a more compact and efficient representation for exploration.

A.4 STRUCTURE VISUALIZATION

B IDENTIFYING SHARED FIELD GROUPS

The identification of `Shared Field Group` nodes is a critical preprocessing step designed to abstract and consolidate recurring schema structures within the database. This process ensures that tables with identical field compositions are linked to a single, canonical group node, thereby reducing redundancy in the graph representation. The methodology is executed in two primary phases:

Table 8: Properties of the `Field` node.

| Property | Description | Example |
|---|---|---|
| `<id>` | Internal unique identifier for the node. | `18` |
| `database` | Name of the parent database. | `NEW_YORK` |
| `description` | A description of the field, if available. | - |
| `name` | The name of the column/field. | `contributing_factor_vehicle_4` |
| `node_type` | Specifies if the field is part of a group or unique to a table. | `unique_field` |
| `sample_data` | Sample data from this column. | - |
| `schema` | Name of the parent schema. | `NEW_YORK` |
| `table` | Name of the parent table (for unique fields). | `NYPD_MV_COLLISIONS` |
| `type` | The data type of the field. | `TEXT` |

Table 9: Statistical overview of the constructed graph representation.

| Component | Count | Percentage |
|---|---|---|
| *Node Types* | | |
| Database | 151 | 0.2% |
| Schema | 267 | 0.3% |
| Table | 7,848 | 8.7% |
| SharedFieldGroup | 542 | 0.6% |
| Field | 81,298 | 90.2% |
| **Total Nodes** | **90,106** | **100.0%** |
| *Relationship Types* | | |
| `HAS_SCHEMA` | 534 | - |
| `HAS_TABLE` | 15,696 | - |
| `USES_FIELD_GROUP` | 11,478 | - |
| `HAS_UNIQUE_FIELD` | 104,808 | - |
| `HAS_FIELD` | 57,788 | - |
| **Total Relationships** | **190,304** | - |

(1) generating a unique signature for each distinct set of fields, and (2) applying a greedy algorithm to select a final, non-overlapping set of shared groups.

**Phase 1: Field Group Signature Generation**    To consistently identify tables that share an identical set of fields, we first compute a content-based signature for the field structure of each table. This signature is derived from the names and data types of all columns within a table.

The process, detailed in Algorithm 1, involves creating a canonical string representation for the set of fields. Each field is formatted as a string concatenation of its name and type (*e.g.*, ``user_id:INTEGER"). To ensure that the signature is independent of the original column order, these formatted strings are sorted alphabetically before being joined into a single, delimited string. Finally, the MD5 hash function is applied to this canonical string to produce a compact and unique 128-bit signature. Any two tables that yield the same signature are considered to have structurally identical schemas.

**Phase 2: Greedy Selection of Non-Overlapping Groups**    After an initial pass over all tables, we obtain a collection of potential shared groups, each identified by its unique signature and associated with a list of tables that match it. A subsequent optimization phase is necessary to produce a final, non-overlapping set of `SharedFieldGroups`. This is crucial because complex schemas may contain nested or overlapping field structures.

---

**Algorithm 1** Field Group Signature Generation

---

**Input:** A table's field set $F_{table} = \{(n_1, t_1), (n_2, t_2), \ldots, (n_k, t_k)\}$, where $n_i$ is a field name and $t_i$ is its data type.
**Output:** A unique signature string $S_{group}$.
1: **function** GENERATESIGNATURE($F_{table}$)
2:     Initialize an empty list $L_{fields}$
3:     **for** each field $(n_i, t_i)$ in $F_{table}$ **do**
4:         $s_i \leftarrow$ Concatenate($n_i$, " : ", $t_i$)             $\triangleright$ Format as "name:type"
5:         Append $s_i$ to $L_{fields}$
6:     **end for**
7:     Sort $L_{fields}$ alphabetically
8:     $S_{canonical} \leftarrow$ Join($L_{fields}$, "|")           $\triangleright$ Create a canonical, delimited string
9:     $S_{group} \leftarrow$ MD5($S_{canonical}$)             $\triangleright$ Compute the MD5 hash
10:    **return** $S_{group}$
11: **end function**

---

We employ a greedy selection algorithm, detailed in Algorithm 2, to resolve this. The core principle is to prioritize the most significant and impactful shared structures. First, all potential groups are sorted in descending order based on two criteria: primarily by the number of tables they encompass, and secondarily by the number of fields they contain (as a tie-breaker). This heuristic prioritizes groups that represent the most widespread schema patterns.

The algorithm then iterates through this sorted list. For each candidate group, it checks if any of its member tables have already been assigned to a previously selected group. If there is no overlap, the group is added to the final set, and all of its member tables are marked as assigned. This process ensures that each table can belong to at most one SharedFieldGroup, resulting in an unambiguous and efficient final graph structure.

---

**Algorithm 2** Greedy Selection of Shared Field Groups

---

**Input:** A collection of all potential groups $G_{all}$, where each group $g \in G_{all}$ has a signature, a set of fields, and a set of matching tables $T_g$.
**Output:** A final, non-overlapping set of shared groups $G_{final}$.
1: **function** SELECTGROUPS($G_{all}$)
2:     $G_{filtered} \leftarrow \{g \in G_{all} \mid |T_g| \geq 2\}$     $\triangleright$ Consider only groups with at least two tables
3:     Sort $G_{filtered}$ in descending order by $|T_g|$, then by number of fields.
4:     Initialize $G_{final} \leftarrow \emptyset$
5:     Initialize set of assigned tables $T_{assigned} \leftarrow \emptyset$
6:     **for** each group $g$ in sorted $G_{filtered}$ **do**
7:         $T_{current} \leftarrow$ the set of tables in group $g$.
8:         **if** $T_{current} \cap T_{assigned} = \emptyset$ **then**     $\triangleright$ Check for overlap with already assigned tables
9:             $G_{final} \leftarrow G_{final} \cup \{g\}$          $\triangleright$ Add group to the final set
10:           $T_{assigned} \leftarrow T_{assigned} \cup T_{current}$        $\triangleright$ Mark tables as assigned
11:         **end if**
12:     **end for**
13:     **return** $G_{final}$
14: **end function**

---

## C   DETAILED ACTION SPACE

This section provides a detailed description of the discrete action space used by the LLM in the exploration stage. At each step of the tree search, the LLM selects one of the following actions to incrementally construct a SQL query based on the current query state and schema context.

- **Select Unused Column:** Adds a new, previously unselected column to the query's 'SELECT' statement. This action progressively expands the breadth of information retrieved by the query.

- **Add Predicate Constraint:** Applies a filter condition to an existing column, typically by adding a 'WHERE' clause. This action narrows the scope of the query, allowing it to focus on specific subsets of data that meet certain criteria.

- **Introduce Join:** Connects the current set of tables to a new table based on foreign key relationships or 'Shared Field Group' linkages. This action is fundamental for exploring relationships across different tables and synthesizing information from multiple sources.

- **Apply Aggregation Function:** Applies a summary function (*e.g.*, 'COUNT', 'SUM', 'AVG', 'MAX', 'MIN') to a previously selected column. This action transforms the query's purpose from simple record retrieval to data summarization, enabling the model to understand the scale and distribution of the data.

- **Add Group By Clause:** Groups the result set by one or more selected non-aggregated columns. This action is typically used in conjunction with aggregation functions to perform categorical analysis, such as calculating metrics for different segments of the data (*e.g.*, "total sales per region").

- **Add Ordering Clause:** Sorts the final result set based on a specified column, either in ascending ('ASC') or descending ('DESC') order via an 'ORDER BY' clause. This enables the discovery of extremes, such as top-performing items or most recent events.

- **Add Having Clause:** Applies a filter condition to the results of a 'GROUP BY' aggregation. Unlike a 'WHERE' clause which filters rows before aggregation, 'HAVING' filters entire groups after aggregation, enabling more complex analytical questions like identifying categories that meet a certain threshold (*e.g.*, "customers with more than 5 orders").

## D ALGORITHM FOR DUAL-AGENT SQL SYNTHESIS

This section provides a detailed algorithmic implementation of the dual-agent framework for SQL synthesis, as described in the Deployment Stage of our methodology. The process is designed as an iterative loop where the **InfoAgent** and **GenAgent** collaborate to refine the context and generate the final SQL query. The InfoAgent is responsible for schema grounding and context management, while the GenAgent leverages the acquired knowledge base to synthesize the query. Algorithm 3 formalizes this collaborative workflow.

**Algorithmic Details**   The algorithm details the iterative process of context refinement and query generation managed by the dual-agent framework. The process begins with an **initialization** phase (Lines 1-4), where the iteration counter, a success flag, and an empty structure for feedback information are prepared. The feedback_info variable is crucial for passing insights from a failed attempt to the next iteration.

Each cycle within the main loop starts with the **InfoAgent** performing schema grounding and expansion (Lines 6-9). It first uses an LLM to extract semantic keywords from the user's utterance and performs a semantic search to retrieve an initial set of schema components. Since this set may be incomplete, a second LLM-driven step expands it by reasoning about logical necessities, such as join keys. Following this, a crucial **context pruning** step occurs (Line 9), where the InfoAgent uses feedback from any previous failed iteration to remove unused schema components, thus narrowing the search space.

The refined context is then passed to the **GenAgent**, which conducts **knowledge retrieval** (Lines 11-12) by using the user's intent to find the most relevant triplets from the knowledge base $\mathcal{K}$. With these triplets as in-context examples, the GenAgent proceeds to **SQL synthesis** (Line 13), generating a candidate query. This query then undergoes **execution and validation** (Lines 15-24). This step has three possible outcomes. An **Execution Failure** (*e.g.*, a syntax error) or a **Semantic Mismatch** (where the query output does not match the user's intent) results in failure feedback being stored in feedback_info for the next loop. A **Success** occurs if the query executes correctly and passes the fidelity check, which sets the success flag and terminates the loop. The entire process continues until a valid query is found or the maximum number of iterations is reached, as defined by the **termination** condition (Lines 27-31), after which the final query or a failure signal is returned.

---

**Algorithm 3** Dual-Agent SQL Synthesis Workflow

---

**Input:** User Utterance $U_{in}$, Database Schema Graph $\mathcal{G}$, Knowledge Base $\mathcal{K}$, Max Iterations $N_{max}$

**Output:** Final SQL Query $Q_{final}$ or failure

1: **function** DUALAGENTSYNTHESIS($U_{in}, \mathcal{G}, \mathcal{K}, N_{max}$)
2:      $i \leftarrow 0$
3:      $is\_success \leftarrow$ **false**
4:      $feedback\_info \leftarrow \emptyset$        ▷ Stores context from failed attempts for pruning
5:      **while** $i < N_{max}$ **and not** $is\_success$ **do**
            ▷ **InfoAgent**: Schema Grounding & Expansion
6:          $keywords \leftarrow$ LLM.ExtractKeywords($U_{in}$)
7:          $S_{initial} \leftarrow$ SemanticSearch($\mathcal{G}, keywords$) ▷ Retrieve initial top-k schema components
8:          $S_{expanded} \leftarrow$ LLM.ExpandContext($U_{in}, S_{initial}$)    ▷ Reason and add necessary related components
9:          $S_{ctx} \leftarrow S_{expanded} \setminus$ PruneUnused($feedback\_info$)      ▷ Refine context based on last failure
           ▷ **GenAgent**: Knowledge Retrieval & SQL Synthesis
10:         $Q_{embed} \leftarrow$ EmbedQueryIntent($U_{in}, S_{ctx}$)
11:         $\mathcal{K}_{retrieved} \leftarrow$ SimilaritySearch($\mathcal{K}, Q_{embed}$)       ▷ Retrieve top-k triplets
12:         $Q_{cand} \leftarrow$ LLM.GenerateSQL($U_{in}, S_{ctx}, \mathcal{K}_{retrieved}$)     ▷ Generate candidate query
          ▷ Execution and Validation Feedback Loop
13:         $result, error \leftarrow$ ExecuteSQL($\mathcal{D}, Q_{cand}$)
14:         **if** $error \neq \emptyset$ **or** $result$ is empty **then**
15:            $feedback\_info \leftarrow (Q_{cand}, S_{ctx},$ "Execution Failed")     ▷ Package feedback for context pruning
16:         **else**
17:            $is\_aligned \leftarrow$ LLM.CheckFidelity($U_{in}, result$)       ▷ Semantic fidelity check
18:            **if** $is\_aligned$ **then**
19:               $Q_{final} \leftarrow Q_{cand}$
20:               $is\_success \leftarrow$ **true**
21:            **else**
22:               $feedback\_info \leftarrow (Q_{cand}, S_{ctx},$ "Semantic Mismatch")
23:            **end if**
24:         **end if**
25:         $i \leftarrow i + 1$
26:      **end while**
27:      **if** $is\_success$ **then**
28:          **return** $Q_{final}$
29:      **else**
30:          **return** failure
31:      **end if**
32: **end function**

---

# E    LLM USAGE

This statement is to disclose the use of large language models (LLMs) in preparing this manuscript, in accordance with ICLR policy. The use of LLMs was strictly limited to improving the language and readability of the text, including grammar correction and stylistic polishing. LLMs did not contribute to any core scientific aspects of this work, such as research ideation, experimental design, or data analysis. All intellectual contributions are the original work of the authors, who assume full responsibility for the final content.

