# OpenReview forum: "SQLAgent: Learning to Explore Before Generating as a Data Engineer"
_ICLR.cc/2026/Conference — ICLR 2026 Conference Withdrawn Submission_

### Official Review · Reviewer_VqYo · 2025-10-18

**Soundness:** 1
**Presentation:** 1
**Contribution:** 2
**Rating:** 2
**Confidence:** 4

**Summary:**

The paper proposes a two‑stage, LLM‑centric framework for Text‑to‑SQL. In the Exploration stage, the system traverses a schema structure (Database→Schema→Table→“Group”→Field) and, via an MCTS‑inspired search, autonomously constructs triplets (S,Q,U). In the Deployment stage, a dual‑agent setup (InfoAgent + GenAgent) retrieves relevant triplets and generates SQL for a user question.

**Strengths:**

S1. Motivation: The paper tackles a real pain point—database‑specific generalization in NL2SQL—by trying to separate knowledge acquisition from generation.

S2. Systematization attempt: The exploration stage aims to build a database‑specific repository of examples that can help downstream generation, which is a reasonable engineering idea even if the theory is thin.

S3. Readable pipeline: The dual‑agent design (InfoAgent/GenAgent) and its runtime loop (Figure 2) are clearly diagrammed, aiding reproducibility at a high level.

**Weaknesses:**

The first set of weaknesses (W1-W5) comes form informal and incorrect database concepts and schema modeling.

W1. Oversimplified/ambiguous “schema” notion: Figure 1 describes a schema as “a logical container for a collection of tables,” but the paper never formalizes essential relational concepts (keys, foreign keys, constraints, views, indices) that are indispensable for reasoning about joins and correctness. This under‑specification leaks into the rest of the method (e.g., join selection and grouping) and undermines the soundness of the search and the triplets that are recorded.

W2. “Group node” concept is ill‑defined and conflated: The left side of Figure 1 says, “When multiple tables share common attributes, use Group Node to perform aggregation.” It is unclear whether “Group” refers to (i) column‑set reuse (the later “Shared Field Group”) or (ii) SQL GROUP BY semantics. The paper uses “group” for both representational abstraction and aggregation, which are orthogonal concepts, producing conceptual confusion.

W3. “Common attributes” not rigorously defined: In practice, “common” is implemented by hashing sorted field names and types to detect identical sets (Appendix B: Algorithm 1). This definition ignores trivial real‑world synonyms/aliases (e.g., course_id vs. c_id), naming conventions, and semantic equivalences. It also treats exact name/type equality as semantic equivalence, which is often false. As a result, the “group” abstraction is brittle and can fail on even mildly heterogeneous schemas.

W4. Unnatural join modeling: The action space explicitly allows joining via “Shared Field Group linkages” (Appendix C). A shared column‑set is not a principled join criterion; joins should be driven by keys/constraints or explicit join conditions. This design invites spurious joins and ill‑formed SQL plans.

W5. From “tree‑like” to graph—but treated as a tree: Section 3.1 calls the representation “tree‑like” while Appendix A formalizes a graph with five node types and several many‑to‑one/many edges (e.g., tables to shared field groups). MCTS on a non‑tree structure requires explicit handling of transpositions/state de‑duplication, which is not specified, compromising search correctness.

W6. The “exploration” search is not soundly specified.
- No clear objective or reward: The MCTS‑inspired procedure removes core MCTS components—no UCT, no explicit reward, and success is largely equated with the query returning a non‑empty result. “Non‑empty” is not a validity metric for semantics; it biases the KB toward easy, high‑cardinality patterns and does not guard against semantically wrong queries that coincidentally return rows.
- Backpropagation on a graph without value semantics: The paper “records positive/negative outcomes on nodes along the path,” but does not define the value being propagated or how it influences future selection beyond a vague “priority.” Without a principled value estimate, the search policy is ad‑hoc.

W7. (S, Q, U) triplets are essentially RAG/few‑shot memory with unclear formalism.
- Definition of S is vague: Section 3.2 defines S as “a subset of the database schema’s structural” (sic). The representation of S, its relation to the actual join path/constraints, and how it ensures semantic alignment are not formalized. The grammatical error here underscores the lack of precision.
- Not clearly different from RAG/few‑shot: The “knowledge base” is a store of (U, Q) (plus a sketch of S). Retrieval of relevant examples at inference time is standard RAG‑style few‑shot prompting. The paper does not articulate a principled difference from prior retrieval‑augmented NL2SQL or “retrieve‑then‑generate” pipelines; it mainly adds an offline synthetic‑example generation step.

W8. The “tree” search is built on a shaky structure; therefore search results aren’t reliable.
Because joins can be proposed via “Shared Field Group linkages” and keys/constraints are not formalized, the state space being searched does not faithfully reflect relational semantics. If the structure is ill‑defined, the resulting triplets and the KB are not reliable, and any downstream search/generation is not sound.

**Questions:**

See the above weaknesses W1-W8.

---

### Official Review · Reviewer_ujZM · 2025-10-28

**Soundness:** 3
**Presentation:** 3
**Contribution:** 2
**Rating:** 4
**Confidence:** 4

**Summary:**

The paper proposes a two-stage framework for Text-to-SQL on complex enterprise databases. Stage 1 (Exploration): autonomously generates a knowledge base of triplets (schema structure, SQL query, natural language) using MCTS-inspired tree search. Stage 2 (Deployment): dual-agent system (InfoAgent + GenAgent) retrieves triplets as in-context examples to generate SQL. Reports 25.78% accuracy on Spider 2.0-Snow vs. 20.84% for ReFoRCE baseline.

**Strengths:**

1. Two-stage design mimics how humans learn databases (explore first, then use)
2. Triplets as executable knowledge vs. static schema descriptions is clever
3. Ablation shows both stages contribute (+5.8 pts for exploration, +5.7 pts for dual-agent). Consistent gains across different LLM backbones (GPT-4o, Claude, Qwen)

**Weaknesses:**

1. Weak baselines
The paper compares against only 2 baselines: (1) ReFoRCE (20.84%) - one concurrent work (2) "Spider-Agent" (12.98%) - a bit naive
Spider 2.0 has a public leaderboard at spider2-sql.github.io. ReFoRCE ranks #8/10 and doesn't seem to be the state-of-the-art. Comparing only against it is misleading.

The paper claims triplets discover "business logic". But: Standard database profiling tools (Great Expectations, dbt) can extract Unique values per column, Cardinality & distributions, FK relationships via value overlap, and Sample data
This is cheaper and deterministic vs. generating 10,000 queries. The paper should show compare against a stronger baseline with Database profiling. Without this, we can't tell if the 6-point gain is from "having examples" or specifically from "having executable triplets."

2. Weak qualitative analysis
The paper shows that accuracy improves but not why:
- No example queries showing when triplets help vs. fail
- No error analysis (what % of failures are wrong joins vs. wrong values?)
- No case studies of successful complex queries
Can't tell if the gain is from better table selection, join inference, or value formatting

3. Exploration cost not justified
Generating triplets requires thousands of query executions during exploration.
But the paper doesn't show the actual wall-clock time for exploration, nor the cost in $ for LLM API calls
It's unclear whether simpler (and cheaper) approaches (profiling + few manual examples) would achieve similar results

**Questions:**

Can you add an ablation with database profiling (column unique values, cardinality, FK detection) as a baseline? This would show whether triplets add value beyond traditional metadata.

Provide 3-4 concrete query examples showing:
- Query that fails without triplets but succeeds with them (and why)
- Query that fails even with triplets (current limitation)
- What triplet was retrieved and how it helped

Error analysis: Of the 74% of queries that still fail, break down failure modes (wrong tables 30%, wrong joins 20%, etc.)

How much does exploration cost in time and $ per database? Is this practical for users?

What makes a good triplet? Are simple queries more useful than complex ones? How many triplets are actually used at deployment?

---

### Official Review · Reviewer_iSCp · 2025-11-03

**Soundness:** 2
**Presentation:** 3
**Contribution:** 2
**Rating:** 4
**Confidence:** 4

**Summary:**

The paper proposes SQLAgent, a two-stage LLM framework for Text-to-SQL. In the Exploration Stage, an LLM guided by a Monte Carlo Tree Search autonomously explores a database schema to create a knowledge base of (schema, SQL, natural language) triplets. In the Deployment Stage, a dual-agent system (InfoAgent and GenAgent) retrieves and iteratively refines SQL generation using these triplets as in-context examples. Experiments on Spider 2.0-Snow show higher execution accuracy than ReFoRCE and Spider-Agent.

**Strengths:**

- Well-engineered framework. The two-stage design is clearly explained and systematically evaluated.
- Strong empirical results. The method achieves higher execution accuracy with detailed ablation and LLM-backbone analysis.
- Good implementation quality. Integration with LangGraph, Neo4j, and FAISS shows solid engineering and reproducibility.

**Weaknesses:**

- Limited conceptual novelty. The “exploration-before-generation” paradigm is well established in prior agentic LLM work (e.g., ReAct, Reflexion, AlphaCode). The contribution mainly adapts this idea to Text-to-SQL.
- Incremental adaptation. The MCTS-style schema exploration and dual-agent refinement are logical extensions rather than fundamentally new ideas.
- Missing qualitative insights. The paper does not show concrete examples illustrating how exploration improves generation.
- Incomplete baseline coverage. Missing recent retrieval- and agent-based systems such as C3, Chess, and Din-SQL.
- Exploration cost unreported. Scalability and runtime overhead are unclear.

**Questions:**

- Clarify the distinction from prior “search-then-generate” or agentic reasoning frameworks.
- Include qualitative examples or visualizations to demonstrate how exploration benefits generation.
- Report exploration runtime, number of generated triplets, and associated cost.
- Expand baselines to include more recent Text-to-SQL models.

---

### Note · Authors · 2026-01-05

I have read and agree with the venue's withdrawal policy on behalf of myself and my co-authors.